# Retracing Phylogenetic, Host and Geographic Origins of Coronaviruses with Coloured Genomic Bootstrap Barcodes: SARS-CoV and SARS-CoV-2 as Case Studies

**DOI:** 10.3390/v15020406

**Published:** 2023-01-31

**Authors:** Alexandre Hassanin, Opale Rambaud

**Affiliations:** Institut de Systématique, Évolution, Biodiversité (ISYEB), Sorbonne Université, CNRS, EPHE, MNHN, UA, 75231 Paris, France

**Keywords:** coronavirus, genome, recombination, COVID-19, reservoir host, secondary host, phylogenetic support, tree reconstruction

## Abstract

Phylogenetic trees of coronaviruses are difficult to interpret because they undergo frequent genomic recombination. Here, we propose a new method, coloured genomic bootstrap (CGB) barcodes, to highlight the polyphyletic origins of human sarbecoviruses and understand their host and geographic origins. The results indicate that SARS-CoV and SARS-CoV-2 contain genomic regions of mixed ancestry originating from horseshoe bat (*Rhinolophus*) viruses. First, different regions of SARS-CoV share exclusive ancestry with five *Rhinolophus* viruses from Southwest China (RfYNLF/31C: 17.9%; RpF46: 3.3%; RspSC2018: 2.0%; Rpe3: 1.3%; RaLYRa11: 1.0%) and 97% of its genome can be related to bat viruses from Yunnan (China), supporting its emergence in the *Rhinolophus* species of this province. Second, different regions of SARS-CoV-2 share exclusive ancestry with eight *Rhinolophus* viruses from Yunnan (RpYN06: 5.8%; RaTG13: 4.8%; RmYN02: 3.8%), Laos (RpBANAL103: 3.3%; RmarBANAL236: 1.7%; RmBANAL52: 1.0%; RmBANAL247: 0.7%), and Cambodia (RshSTT200: 2.3%), and 98% of its genome can be related to bat viruses from northern Laos and Yunnan, supporting its emergence in the *Rhinolophus* species of this region. Although CGB barcodes are very useful in retracing the origins of human sarbecoviruses, further investigations are needed to better take into account the diversity of coronaviruses in bats from Cambodia, Laos, Myanmar, Thailand and Vietnam.

## 1. Introduction

Tree reconstruction methods, such as Bayesian inference and maximum likelihood (ML), are very popular in deciphering phylogenetic relationships between pathogens based on multiple sequence alignments. In particular, newly discovered coronaviruses are routinely described based on Bayesian and ML trees reconstructed from whole or partial genome alignments [1,2,3,4]. For instance, most recent studies on SARS-CoV-2, the virus involved in the COVID-19 pandemic, have published a whole-genome tree of the subgenus *Sarbecovirus* (family Coronaviridae, genus *Betacoronavirus*) in which the human virus was found to be closely related to RaTG13, a virus detected in a horseshoe bat of the species *Rhinolophus affinis* sampled in 2013 in the Yunnan province of China [4,5,6,7]. However, discordant placements of SARS-CoV-2 were supported in phylogenetic trees based on different genomic fragments. This was well illustrated in Zhou et al. [7], in which SARS-CoV-2 appeared closely related to four bat viruses, namely RaTG13, RmYN02, RpYN06 and RshSTT200, in the RNA-dependent RNA polymerase (*RdRp*) gene tree, sister-group of RmYN02 and RpYN06 in the ORF1ab tree, and linked to RaTG13 in the *Spike* gene tree. Such conflicting results between gene trees are typically explained by genomic recombination, a process resulting in mosaic genomes containing regions from different parental viruses. The most widely-accepted model to explain recombination in coronaviruses is the copy-choice model, also named the template switching model: during RNA replication, the viral RdRp can pause on the RNA template and switch to another template, thereby generating a recombinant RNA molecule with mixed ancestry [8]. Our recent study has also suggested that circular RNAs may be involved in the process of genomic recombination [9].

Several previous studies have provided strong evidence that *Sarbecovirus* genomes are derived from a large number of past recombination events in bats [9,10] and also in humans [11,12]. This means that each *Sarbecovirus* RNA genome has a specific mosaic structure, i.e., a unique combination of genomic fragments showing different evolutionary histories: some fragments may be shared with only one virus, suggesting recent ancestry; other fragments may provide only support for grouping with several divergent viruses, suggesting older ancestry (or insufficient sampling of viruses); the origin of some fragments may be very difficult to interpret due to multiple recombination events in overlapping or nested genomic locations and loss of phylogenetic signal over time (multiple nucleotide substitutions at the same site). To better interpret conflicting phylogenetic signals due to genomic recombination, we report hereinafter a new approach, coloured genomic bootstrap (CGB) barcodes, in which a virus genome of special interest (e.g., the common ancestor of SARS-CoV-2) is represented by a succession of coloured regions showing the best phylogenetic signals, i.e., including the fewest number of closest relatives among available viral genomes. The method was applied to an alignment of 75 *Sarbecovirus* genomes to provide new insights into the phylogenetic, host and geographic origins of SARS-CoV-2 and SARS-CoV (virus involved in the 2002–2003 and 2003–2004 SARS outbreaks [13,14]).

## 2. Materials and Methods

### 2.1. Nucleotide Alignment of Sarbecovirus genomes

Complete genomes available for *Sarbecovirus* in June 2022 in GenBank (https://www.ncbi.nlm.nih.gov/, accessed on 15 June 2022), GISAID (https://www.epicov.org/, accessed on 15 June 2022), and NGDC (https://ngdc.cncb.ac.cn/, accessed on 15 June 2022) databases were downloaded in Fasta format. Sequences with large amount of missing data were removed. Several genomes showing perfect identity or high nucleotide similarity (more than 99.9% of nucleotide identity) were published for pangolin sarbecoviruses from Guangxi (5 sequences), bat sarbecoviruses from Thailand (5 sequences), Cambodia (two sequences), etc. For these clusters, a single genome was retained for our analyses. The international databases contain millions of SARS-CoV-2 genomes and hundreds of SARS-CoV genomes. For human SARS-CoV-2, we decided to include in the alignment the reference genome and one representative for each of the six variants of concern (VOC; Alpha, Beta, Gamma, Delta, Epsilon, Omicron), which were selected under NCBI Virus (ncbi.nlm.nih.gov/labs/virus/, accessed on 15 June 2022) using the following criteria: country for which the highest number of sequences was available (i.e., USA); Illumina sequencing; no stop codon in the coding sequences (cds); and no missing data. We also included two SARS-CoV-2 genomes extracted from small carnivores of the family Mustelidae, i.e., *Mustela lutreola* (European mink) and *Neovison vison* (American mink), differing by more than 0.1%. For human SARS-CoV, we included in the alignment four genomes showing more than 0.1% of nucleotide divergence. Similarly, we included three SARS-CoV-like genomes extracted from *Paguma larvata* (masked palm civet), as this small carnivore of the family Viverridae was identified as a possible intermediate host between bats and humans during the 2002–2003 and 2003–2004 SARS outbreaks [13,14]. The details on the 75 selected genomes are provided in Appendix A. They include all viral lineages previously described within the subgenus *Sarbecovirus* [4,5,6,7,15,16].

The nucleotide sequences were aligned in Geneious Prime^®^ 2020.0.3 with MAFFT version 7.450 [17] using default parameters. Then, the alignment was corrected manually on AliView 1.26 [18] based on nucleotide and amino-acid sequences using the three following criteria: (i) the number of indels was minimized because they are rarer events than nucleotide substitutions; (ii) transitions were privileged over transversions because they are more frequent; and (iii) changes between similar amino-acids (as shown by the ClustalX colour scheme) were preferred. The insertions found in only one virus were removed from the whole-genome alignment.

### 2.2. Phylogenetic Analyses

Maximum likelihood analysis of the whole-genome alignment of sarbecoviruses was carried out using RAxML 8.2.11 [19], different GTR+G models for the three codon-positions and non-coding regions, and 1000 bootstrap replicates. The RAxML bootstrap trees were executed in PAUP* version 4.0a [20] to construct the bootstrap 50% majority-rule consensus tree.

To examine the distribution of phylogenetic support along the whole-genome alignment, the dataset was bootstrapped under the SWB program [9] using a window of W nucleotides (W parameter) moving in steps of 50 nt (S parameter). The window size (W) is a key parameter for SWB analyses because the amount of phylogenetic signal depends on both the number of nucleotide sites and their evolutionary rates [9]. For that reason, we decided to perform five SWB analyses with the same step parameter (S) of 50 nt but using five different window sizes, i.e., 400 nt, 500 nt, 600 nt, 1000 nt and 2000 nt (Table 1). The smallest window size (W = 400 nt) was applied to detect possible changes in phylogenetic relationships due to the recombinant origin of small genomic regions, whereas the largest window size (W = 2000 nt) was used to guarantee enough phylogenetic signal (informative sites) for bootstrap analyses. The intermediate window sizes (500, 600 and 1000 nt) were used to better interpret the differences between SWB results based on the two extreme values. In the SWB program, each window bootstrap (WB) sub-dataset was automatically run in RAxML [19] with a GTR+G model and 100 bootstrap replicates. The SWB output is a CSV file containing the window bootstrap percentages (WBP) calculated for each WB sub-datasets and for all the bipartitions (nodes) reconstructed during the SWB analysis. For example, the SWB_400_ output (SWB analysis based on a window of 400 nt) includes 1,213,380,595 WBP values (595 WB sub-datasets × 204,001 bipartitions), whereas the SWB_2000_ output includes 11,189,625 WBP values (563 WB sub-datasets × 19,875 bipartitions) (Table 1).

The bootstrap bipartitions generated from the five SWB analyses based on different window sizes were used for SuperTRI analyses [21] to reconstruct the trees showing the most reliable phylogenetic relationships. The LFG program [9] was used to convert the SWB output file into bootstrap log files, which were then used as inputs in SuperTRI v57 [21] to construct a MRP (Matrix Representation with Parsimony) file. For example, the SWB_400_ output was converted with the LFG program into 595 bootstrap log files, and these files were further transformed into a MRP file using SuperTRI v57. In the MRP_400_ file, each of the 750,330 characters represents one SWB bipartition with its assigned bootstrap percentage calculated in one of the 595 window bootstrap analyses. The MRP_400_ file was then executed in PAUP* version 4.0a [20] using 1000 bootstrap replicates of weighted parsimony (with WBPs of the SWB_400_ analysis assigned as weights) in order to reconstruct the SuperTRI bootstrap 50%-majority-rule consensus (SB_400_) tree. Finally, the mean window bootstrap percentages (MBP_400_) were calculated automatically in SuperTRI v57 [21] for all nodes of the SB_400_ trees. The same approach was conducted using the SWB outputs obtained with the sliding windows of 500, 600, 1000 and 2000 nt. In total, we therefore reconstructed five SB trees with MBP values at the nodes.

### 2.3. Construction of Genomic Bootsrap Barcodes

The BBC program [9] was used to select only SWB bipartitions (i.e., phylogenetic hypotheses) showing one or more WBP values ≥ 50% and to construct their corresponding genomic bootstrap (GB) barcodes. For example, the SWB_400_ output contains 204,001 bipartitions, each with 595 WBP values. After BBC analysis, the GB_400_ barcodes were done for the 1263 selected SWB_400_ bipartitions showing one or more WBP ≥ 50% (Table 1). A GB barcode is a small image representing the genome alignment and in which the n WBP values (n = 595 with an alignment of 30,115 nt and a window size of 400 nt) obtained for the BBC bipartition of interest were transformed into coloured vertical bars using the following code: green for WBP ≥ 70%; grey for 30% < WBP < 70%; and red for WBP ≤ 30%. Using the same BBC procedure but a different SWB output (either SWB_500_, SWB_600_, SWB_1000_ or SWB_2000_), we constructed 1240 GB_500_ barcodes, 1212 GB_600_ barcodes, 1061 GB_1000_ barcodes, and 818 GB_2000_ barcodes (Table 1). All the 5594 GB barcodes and the five BBC output files (BBC_400_, BBC_500_, BBC_600_, BBC_1000_, and BBC_2000_) were deposited in the Open Science Framework (OSF) platform at https://osf.io/nj57e/, accessed on 15 June 2022.

### 2.4. Construction of Coloured Genomic Bootstrap Barcodes

A new method was developed in this study for constructing coloured genomic bootstrap (CGB) barcodes for a virus of special interest or the common ancestor of several viruses. A phylogenetic CGB barcode is a small image representing the genome of a virus (or an ancestral virus) in which the different colours show the best phylogenetic signals, i.e., containing the fewest number of closely-related viruses, detected in the different regions of the genomic alignment used for the analyses. Two other kinds of CGB barcodes were derived from the phylogenetic CGB barcodes: the host CGB barcodes showing the host origins of the closely-related viruses and the geographic CGB barcodes showing the geographic origins of the closely-related viruses. For this study, we choose to reconstruct the CGB barcodes for the common ancestor of seven selected SARS-CoV genomes and for the common ancestor of nine selected SARS-CoV-2 genomes (see Appendix A for the origin of the sequences used in this study). To avoid repetitions, only the SARS-CoV-2 procedure is described below.

In the first step, the five BBC output files were imported in Microsoft^®^ Excel. They represent five lists of BBC bipartitions, i.e., SWB bipartitions showing at least one WBP ≥ 50% (Table 1). They contain the following information for each BBC bipartition: bipartition number, lists of viruses included in the bipartition, binary representations of the bipartition (*: viruses included in the bipartition; -: viruses excluded from the bipartition), WBPs obtained for the n sliding window bootstrap analyses (e.g., n = 595 for SWB_400_ and n = 563 for SWB_2000_). The five BBC outputs include the following number of BBC bipartitions: 1263 for BBC_400_, 1240 for BBC_500_, 1212 for BBC_600_, 1061 for BBC_1000_, and 818 for BBC_2000_ (Table 1). The outputs were entered sequentially into the same Excel file starting with BBC_400_ and ending with BBC_2000_ and highlighted with five different background colours. To allow comparisons between the five BBC results, W and S parameters (window size and moving steps) were used to calculate the median positions (pos.) for the n WBPs calculated in each of the five SWB analyses.

In the second step, only BBC bipartitions including all the nine SARS-CoV-2 sequences were selected: 178 bipartitions for BBC_400_, 176 bipartitions for BBC_500_, 175 bipartitions for BBC_600_, 163 bipartitions for BBC_1000_ and 132 bipartitions for BBC_2000_ (Table 1). For each of the five BBC lists, the single bipartition including only SARS-CoV-2 sequences was removed. Then, other BBC bipartitions were ranked in increasing order of size, from +1 (for the bipartitions including the nine SARS-CoV-2 sequences + one closely-related virus) to +66 (=75 − 9, for the single bipartition including all viruses of our dataset). To make comparisons between WBPs calculated in the five SWB analyses, a new column was inserted to renumber BBC_500_, BBC _600_, BBC_1000_, and BBC_2000_ bipartitions using BBC_400_ numbers as references.

In the third step, all WBPs ≥ 70% were highlighted in green and all WBPs comprised between 50% and 70% were highlighted in yellow green using conditional formatting options in Microsoft^®^ Excel. We performed the comparisons starting with bipartitions +1, i.e., containing all SARS-CoV-2 sequences and another virus. Due to past events of genomic recombination, we found several bipartitions +1 supporting conflicting phylogenetic relationships. For each of these BBC bipartitions, we identified the intervals of genomic regions containing a robust phylogenetic signal (GRPS) using the criteria developed below. Then we proceeded similarly by analysing other genomic fragments for bipartitions +2 (containing all SARS-CoV-2 sequences and two other viruses), bipartitions +3 (containing all SARS-CoV-2 sequences and three other viruses), etc. In this way, we were able to identify the closest virus(es) to SARS-CoV-2 in all regions of our genome alignment. All selected GRPS contained a robust phylogenetic signal (WBP ≥ 70%) in at least two WB sub-datasets of the BBC_400_, BBC_500_ or BBC_600_ results. The 5’ and 3’ ends of GRPS were extended using the following criteria: (i) by accepting WBPs between 50% and 70% for BBC_400_ results; (ii) when median positions showed an average WBP ≥ 50% for BBC_400_, BBC_500_ and BBC_600_ results; and (iii) when median positions showed an average WBP ≥ 50% for all the five BBC results (BBC_400_, BBC_500_, BBC_600_, BBC_1000_, and BBC_2000_). This strategy was adopted for three major reasons: (i) GRPS of small lengths cannot be detected using SWB analyses based on the largest window sizes (i.e., 1000 nt and 2000 nt); (ii) due to stochastic variation in bootstrap values, the comparisons between the three SWB analyses based on the smallest window sizes (i.e., 400 nt, 500 nt and 600 nt) allow us to better detect GRPS of small lengths; (iii) GRPS of large size can be erroneously interrupted if we consider only SWB analyses based on the smallest window sizes because they contain lesser amounts of phylogenetic signal [9]. It is therefore important also to make comparisons with BBC results based on the largest window sizes (i.e., 1000 nt and 2000 nt).

In the fourth step, the intervals of GRPS (5′ and 3′ median positions in the whole-genome alignment) were written in a new CSV file for each of the BBC bipartitions including SARS-CoV2 sequences in which one or more GRPS were identified as the best phylogenetic signals (i.e., containing the fewest number of closely-related viruses). A specific colour code was chosen for each of the selected BBC bipartitions and the file was used as input in the CGB program (python script) to construct phylogenetic CGB barcodes of different colours. Two other files were derived from the original CSV file: (i) a file for host CGB barcodes, in which different colours were assigned to the BBC bipartitions including viruses extracted from the following nine taxa: *Rhinolophus affinis*, *Rhinolophus malayanus*, *Rhinolophus marshalli*, *Rhinolophus pusillus*, *Rhinolophus shameli*, *R. affinis* + *Manis javanica*, *Rhinolophus* species, *Rhinolophus* species + *M. javanica*, bat species + *M. javanica*; and (ii) a file for geographic CGB barcodes, in which different colours were chosen for the BBC bipartitions including viruses collected in the following nine geographic areas: Cambodia, North Laos, SE Asia, Yunnan, North Laos + Yunnan, SE Asia + Yunnan, North Laos + China, SE Asia + China, and SE Asia + China + Japan. All files used to construct the CGB barcodes for SARS-CoV-2 and SARS-CoV (including BBC_400_, BBC_500_, BBC_600_, BBC_1000_, BBC_2000_ files, the two Excel files used to identify GRPS intervals, and the six CSV files used for phylogenetic, host and geographic CGB barcodes) were deposited at https://osf.io/nj57e/, accessed on 15 June 2022.

## 3. Results

### 3.1. Phylogenetic Analyses Based on An Alignment of 75 Sarbecovirus genomes

In this study, 75 genomes of the subgenus *Sarbecovirus* (Appendix A) were aligned to infer phylogenetic relationships. The positions of the coding sequences were the following in our final alignment of 30,115 nucleotides (nt): 256–21,654 for ORF1ab, including the RNA-dependent RNA polymerase gene [*RdRp*] at positions 13,540–16,335; 21,664–25,557 for the spike (*S*) gene; 25,567–26,394 for ORF3a; 26,419–26,649 for the envelope (*E*) gene; 26,704–27,375 for the membrane (*M*) gene; 27,388–27,579 for ORF6; 27,589–28,101 for ORF7ab; 28,108–28,485 for ORF8; 28,504–29,775 for the nucleocapsid (*N*) gene; and 29,803–29,919 for ORF10 (Figure 1).

Two different approaches were used for phylogenetic reconstruction: whole-genome bootstrap (WGB) analysis versus SuperTRI bootstrap (SB) analysis. On the one hand, a classical ML approach was applied with the RAxML method using the whole-genome alignment and 1000 bootstrap replicates. The WGB 50% majority-rule consensus tree is shown in Figure 2. On the other hand, five SWB analyses were conducted using five different window sizes (400, 500, 600, 1000 and 2000 nt) and the results were used to reconstruct five SuperTRI bootstrap (SB) consensus trees (with MBP values indicated at the nodes). The five SB trees (available at https://osf.io/nj57e/, accessed on 15 June 2022) were found to be very similar, except for a few nodes. For instance, Rs4237 and Rs4247 are sister viruses in SB_400_, SB_500_, SB_600_ trees, whereas Rs4247 is more closely related to Rs4081 in SB_1000_ and SB_2000_ trees. Additionally, MBP values were generally lowest for SB_400_ nodes and highest for SB_2000_ nodes. Therefore, we only reported MBP_400_ and MBP_2000_ values at the nodes of the WGB tree of Figure 2. The MBP values written in black indicate nodes recovered as monophyletic in all the five SB trees, whereas the MBP values highlighted in red indicate nodes not found monophyletic in SB trees. In Figure 2, we also showed the GB barcodes built from SWB_400_ and SWB_2000_ analyses for all nodes of the WGB tree supported by BP_WG_ ≥ 90% or recovered as monophyletic in SB trees.

In the WGB tree of Figure 2, 48 out of the total 69 nodes (70%) are supported by BP_WG_ ≥ 90%. Robustness comparisons with SB trees revealed that 24 of these 48 robust nodes (50%) are associated with MBP_400/2000_ ≤ 30%, indicating that the phylogenetic signal is restricted to only one or several regions of the genome alignment, representing in total less than 30% of the whole genome alignment [9]. For examples, the monophyly of *RecSar*, which includes the three bat viruses RpPrC31, RsVZC45, and RsVZXC21, was supported by MBP_400_ = 15 and MBP_2000_ = 21; the sister-group relationship between SARS-CoV and Rs4231 + Rs4874 was supported by MBP_400_ = 3 and MBP_2000_ = 8; and the grouping of SARS-CoV-2 with RaTG13, RmBANAL52, RmaBANAL236, and RpBANAL103 was supported by MBP_400_ = 2 and MBP_2000_ = 4. Note that the later node was not recovered monophyletic in the five SB trees reconstructed from SWB analyses (MBP values written in red in Figure 2). For all these nodes, the GB barcodes revealed that the phylogenetic signal is restricted to one or several small genomic regions (in green, WBP ≥ 70%), whereas most other genomic regions (in red, WBP ≤ 30%) can support different phylogenetic relationships, as illustrated with CGB barcodes described in Section 3.2 and Section 3.3.

Seventeen of the 48 robust nodes (35%; BP_WG_ ≥ 90%) of the WGB tree of Figure 2 are supported by MBP_400/2000_ comprised of between 30% and 70%, indicating that the phylogenetic signal covers many regions or one large region of the genome alignment. A first example concerns the node supporting the monophyly of SARS-CoV related coronaviruses (*SCoVrC*; MBP_400/2000_ = 36/44), a large group containing four human SARS-CoV, three civet SARS-CoV, and 37 bat viruses (written in black in Figure 2). A second example is the node supported by MBP_400/2000_ = 30/45, which includes five viral lineages showing different synonymous nucleotide composition [16] (highlighted by different colours in Figure 2): (i) SARS-CoV-2 related coronaviruses (*SCoV2rC*, in green); (ii) the three viruses showing evidence of genomic recombination between *SCoVrC* and *SCoV2rC* (*RecSar*, in dark blue); (iii) the two pangolin sarbecoviruses (in light blue); (iv) the bat sarbecoviruses from Yunnan showing a very divergent synonymous nucleotide composition (*YunSar*, in red); and (v) the Rco319 virus from Japan (in orange).

Only seven of the 48 robust nodes (15%; BP_WG_ ≥ 90%) of the WGB tree of Figure 2 are supported by MBP_400/2000_ ≥ 70%, indicating that the phylogenetic signal is present almost everywhere in the genome alignment, as shown by GB barcodes, which are green (supported by WBP ≥ 70%) in most regions (GB_400_) or all regions (most GB_2000_ barcodes) of the alignment. They include (i) the monophyly of SARS-CoV, which contains four human sequences and three civet sequences (MBP_400/2000_ = 74/100); (ii) the monophyly of SARS-CoV-2, which contains seven human sequences and two mink sequences (MBP_400/2000_ = 82/99); (iii) the monophyly of *YunSar*, which is represented by two bat sarbecoviruses from Yunnan, Ra7909 and RstYN04 (MBP_400/2000_ = 99/100); (iv) the sister-group relationship between RhGB01 and RhKhosta2 (MBP_400/2000_ = 85/99); (v) the sister-group relationship between Rmac1 and Rmac279 (MBP_400/2000_ = 93/100); (vi) the sister-group relationship between Rf1 and Rf273 (MBP_400/2000_ = 80/85); and (vii) their grouping with RfJiyuan84 (MBP_400/2000_ = 82/94). The nine SARS-CoV-2 genomes were sampled between December 2019 (HsRef, GenBank: NC_045512) and July 2022 (HsOmicron, GenBank: OP010674) and we know that their most recent common ancestor (MRCA) emerged a few weeks before the end of 2019. Due to their recent divergence, these viruses have very similar genomes. Indeed, pairwise nucleotide distances are comprised between 0.04% and 0.28%. We also found very similar distances for most other closely-related viruses supported by high MBP values: between 0.11% and 0.41% for the seven SARS-CoV genomes; 0.46% between Rmac1 and Rmac279; 0.76% between Rf1 and Rf273. All these pairwise distances are the smallest calculated for our dataset. In contrast, the two other virus pairs supported by MBP_400/2000_ ≥ 70% were found to be more distant: 2.0% between Ra7909 and RstYN04, and 12.5% between RhGB01 and RhKhosta2.

Interrelationships between the seven human and two mink SARS-CoV-2 sequences were not robust in the WGB tree of Figure 2, except the basal position of HsRef (BP_WG_ = 92) which was supported by one GRPS (pos. 14,251–14,750, with an exclusive Uracil in pos. 14,507). However, we found discordant SARS-CoV-2 relationships. For instance, HsGamma and HsOmicron were related with HsDelta based on pos. 9851–10,350 (including an exclusive Uracil in pos. 10,127) or with HsAlpha based on pos. 28,801–29,350 (including an exclusive Cytosine in pos. 29,116).

Interrelationships between the four human and three civet SARS-CoV sequences were not robust in the WGB tree of Figure 2, except three nodes: (i) the sister-group relationship between PlSZ61 and PlGZ81 (BP_WG_ = 100), which was supported by 19 GRPS (representing 52% of the WG alignment) containing six exclusive nucleotides (pos. 4270: A; pos. 9503: C; pos. 18,349: C; pos. 24,083: U; pos. 25,640: A; pos. 27,499: G); (ii) the monophyly of civet SARS-CoV (PlSZ3, PlSZ61, and PlGZ81; BP_WG_ = 94), which was supported by two GRPS (pos. 25,001–26,200 and 26,401–26,950; 6% of the alignment) containing an exclusive Guanine in pos. 25,276; and (iii) the grouping of HsRef with HsShangaiQXC1 (BP_WG_ = 100), which was supported by 10 GRPS (26% of the alignment) containing three exclusive nucleotides (pos. 9647: U; pos. 23,497: G; pos. 28,156: U). By comparison, the monophyly of human SARS-CoV (BP_WG_ = 52) was supported by two GRPS (pos. 22,651–24,200 and 25,251–25,750; 7% of the alignment) containing an exclusive Uracil in pos. 25,585. However, we found some GRPS supporting discordant relationships. In particular, the monophyly of human SARS-CoVs was contradicted by one GRPS supporting the grouping of the three civet SARS-CoVs with HsRef and HsShanghaiQXC1 (pos. 25,801–26,550; containing an exclusive Adenine in pos. 26,143), one GRPS supporting the grouping of PlSZ3 with HsRef, HsShanghaiQXC1, and HsGD0 (pos. 2451–2950; containing an exclusive Guanine in pos. 12,750), as well as two GRPS supporting the grouping of PlSZ61 and PlGZ81 with HsRef, HsShanghaiQXC1, and HsGD0 (pos. 17,651–18,100 and 20,901–21,400; containing an exclusive Guanine in pos. 21,161).

### 3.2. Coloured Genomic Bootstrap Barcodes Reconstructed for the Ancestor of SARS-CoV

We found that the best phylogenetic signals detected for the common ancestor of SARS-CoV involved 80 GRPS and 49 BBC bipartitions. The file used to identify the intervals of the 80 GRPS (5′ and 3′ median positions in the whole-genome alignment) is available at https://osf.io/nj57e/ (accessed on 15 June 2022).

The phylogenetic, host and geographic CGB barcodes constructed for the common ancestor of SARS-CoV are shown in Figure 3. The phylogenetic CGB barcodes indicate that 25.6% of the SARS-CoV genome shares exclusive ancestry with five *Rhinolophus* viruses (bipartitions +1 in Figure 3) detected in three provinces of Southwest China: three viruses from Yunnan, including RfYNLF/31C sampled in *R. ferrumequinum* (eight GRPS representing 17.9% of the WG alignment), RpF46 sampled in *R. pusillus* (two GRPS; 3.3% of the alignment); RaLYRa11 sampled in *R. affinis* (one GRPS; 1.0% of the alignment); one virus from Guangxi, Rpe3 sampled in *R. pearsonii* (one GRPS; 1.3% of the alignment); and one virus from Sichuan, RspSC2018, sampled in an unidentified species of *Rhinolophus* (two GRPS; 2.0% of the alignment).

If we consider the BBC bipartitions uniting SARS-CoV with two to five closely-related viruses (n = +2, +3, +4 or +5 in Figure 3), they involve eight additional viruses, all found in *Rhinolophus* bats collected in Yunnan: RmYN07, Rs4231, Rs4237, Rs4247, Rs4874, Rs7327, Rs9401, and RstYN09. These results therefore suggest that SARS-CoV originated from horseshoe bat (genus *Rhinolophus*) viruses. This hypothesis was confirmed with our analyses of Figure 4B, which revealed that 100% of the phylogenetic CGB barcodes reconstructed for SARS-CoV involved *Rhinolophus* viruses. As shown in Figure 4A, the most important contributors are five sarbecoviruses extracted from horseshoe bats: RfYNLF/31C (which is 65% involved), Rs7327 (63%), Rs9401 (63%), Rs4874 (62%), and Rs4084 (61%). Importantly, all the 22 viruses showing a significant contribution in SARS-CoV CGB barcodes (≥25%) belong to the *SCoVrC* lineage.

Several regions of the SARS-CoV genome were found to be exclusively related to RfYNLF/31C, a virus found in *Rhinolophus ferrumequinum*. These regions represent 17.9% of the whole-genome alignment and include several dispersed fragments in ORF1ab, including two fragments in the *RdRp* gene (RfYNLF/31C; pos. 13,301–13,900 and 14,451–15,300), as well as almost the complete ORF7a (RfYNLF/31C; pos. 27,601–27,950). The genomic contribution of the other species is much lower (≤ 3.3%), except for *Rhinolophus sinicus* (13.1%), as most parts of the *Spike* gene of SARS-CoV were found to be closely related to viruses detected in this species (Rs4231 + Rs4874 in pos. 21,451–22,700; Rs7327 + Rs9401 in pos. 24,151–25,400; and Rs3367 + Rs4084 + Rs4874 + RsSHC014 + Rs7327 + Rs9401 in pos. 23,751–24,150 and 25,401–25,850). Note however that a small fragment of the *Spike* gene was found to be linked to a virus detected in *R. affinis* (RaLYRa11; pos. 22,801–23,100). Overall, the results presented in Figure 4B show that the three *Rhinolophus* species with the highest contribution to the phylogenetic CGB barcodes of SARS-CoV are *R. ferrumequinum* (76%), *R. sinicus* (71%) and *R. affinis* (58%).

The geographic CGB barcodes indicate that most regions of the SARS-CoV genome are exclusively related to bat viruses from Southwest China (84%; highlighted in orange, yellow and red in Figure 3), including Yunnan (41%), Sichuan (2%), and Guangxi (1%). Consistent with this, we found that 100% of the phylogenetic CGB barcodes reconstructed for SARS-CoV involved viruses detected in Southwest China, and the contribution of the Yunnan province is 97%, which is much more important than that of the three other provinces of Southwest China, i.e., Guizhou (53%), Guangxi (46%), and Sichuan (33%) (Figure 4C).

### 3.3. Coloured Genomic Bootstrap Barcodes Reconstructed for the Ancestor of SARS-CoV-2

We found that the best phylogenetic signals detected for the common ancestor of SARS-CoV-2 involved 95 GRPS and 52 BBC bipartitions. The file used to identify the intervals of the 95 GRPS (5′ and 3′ median positions in the whole-genome alignment) is available at https://osf.io/nj57e/ (accessed on 15 June 2022).

The phylogenetic, host and geographic CGB barcodes constructed for the common ancestor of SARS-Co2 are shown in Figure 5. The phylogenetic CGB barcodes indicate that 23.4% of SARS-CoV-2 genome shares exclusive ancestry with eight *Rhinolophus* viruses (bipartitions +1 in Figure 5): three from Yunnan, RaTG13 sampled in *R. affinis* (four GRPS representing 4.8% of the alignment), RmYN02 sampled in *R. malayanus* (two GRPS; 3.8% of the alignment), and RpYN06 sampled in *R. pusillus* (two GRPS; 5.8% of the alignment); four from northern Laos, RpBANAL103 sampled in *R. pusillus* (one GRPS; 3.3% of the alignment), RmaBANAL236 sampled in *R. marshalli* (one GRPS; 1.7% of the alignment), RmBANAL52 (one GRPS; 1% of the alignment) and RmBANAL247 (one GRPS; 0.7% of the alignment), both sampled in *R. malayanus*; and one from northern Cambodia, RshSTT200 sampled in *R. shameli* (two GRPS; 2.3% of the alignment). The BBC bipartitions including between two and five additional viruses (n = +2, +3, +4 or +5 in Figure 5) involve only two additional viruses: one isolated from the Sunda pangolin (*M. javanica*), named MjGuangxi, and another found in *Rhinolophus acuminatus* from Eastern Thailand, named RacCS203. These results therefore suggest that SARS-CoV-2 originated from horseshoe bat (genus *Rhinolophus*) viruses rather than pangolin viruses. This hypothesis was confirmed with the results of Figure 6B, which show that 100% of the phylogenetic CGB barcodes reconstructed for SARS-CoV-2 involved *Rhinolophus* viruses, whereas the contribution of *Manis* viruses remains modest (24%). As shown in Figure 6A, the most important contributors are six viruses extracted from horseshoe bats of northern Laos and Yunnan: RpBANAL103 (which is 72% involved), RmBANAL52 (70%), RpYN06 (65%), RmaBANAL236 (65%), RmYN02 (63%), and RaTG13 (60%). Importantly, all the 10 viruses showing a significant contribution (≥25%) belong to the *SCoV2rC* lineage (between 72% and 35%), except RpPrC31 (31%), which belongs to the *RecSar* group [9,16].

Several regions of the SARS-CoV-2 genome were found to be exclusively related to viruses extracted from *R. malayanus* (RmYN02, RmBANAL52 and RmBANAL247; representing 6.8% of the whole-genome alignment), *R. pusillus* (RpYN06 and RpBANAL103; 9.1%), *R. affinis* (only RaTG13; 4.8%), *R. shameli* (only RshSTT200; 2.3%), and *R. marshalli* (only RmaBANAL236; 1.7%). In particular, the *Spike* gene of SARS-CoV-2 can be related to genomic fragments from viruses of different species, including *R. affinis* (RaTG13; pos. 21,701–22,400), *R. affinis* + *M. javanica* (RaTG13 + MjGuangxi; pos. 22,501–22,700), *R. shameli* (RshSTT200; pos. 24,001–24,250 and 25,401–25,850), and *R. malayanus* (RmYN02 + RmBANAL247; pos. 24,951–25,350). The results presented in Figure 6B show that the four *Rhinolophus* species with the highest contribution to the phylogenetic CGB barcodes of SARS-CoV-2 are *R. pusillus* (82%), *R. malayanus* (81%), *R. marshalli* (65%) and *R. affinis* (60%).

The geographic CGB barcodes indicate that most regions of the SARS-CoV-2 genome are exclusively related to bat viruses from Yunnan and Southeast Asia (81%), some of them being specific to Yunnan (17% highlighted in red in Figure 5) or Southeast Asia (19%; northern Laos representing 16%, in yellow). Consistent with this, we found that 100% of the phylogenetic CGB barcodes reconstructed for SARS-CoV-2 involved viruses detected in Southeast Asia and Yunnan (Figure 6C). However, sampling efforts were much more important in Yunnan: 53% of the *Sarbecovirus* genomes used in our study were collected in Yunnan against 7% for those found in Laos. The contribution of bat sarbecoviruses from Yunnan is 81%. The contribution of bat sarbecoviruses from Laos is 78%, which is much more important than the contribution of bat sarbecoviruses found in other Southeast Asian countries: 47% for the virus from Cambodia and 35% for the virus from Thailand (Figure 6C).

## 4. Discussion

### 4.1. Phylogenetic Trees versus Phylogenetic CGB Barcodes

One or several phylogenetic trees have been published in all reports dealing with the origin and evolution of SARS-CoV-2 [4,5,6,7,15,16]. In these studies, SARS-CoV-2 was generally found to be closely related to RaTG13, a virus detected in a *R. affinis* bat collected in Yunnan in 2013 [4,5,6,7]. However, phylogenetic analyses based on different genomic regions were found to support conflicting relationships [6,7,9,16]. This was well illustrated in the Figure 3 of Zhou et al. [7], in which SARS-CoV-2 appeared closely related to RaTG13, RmYN02, RpYN06, and RshSTT200 in the *RdRp* gene tree (BP = 94), sister to RmYN02 and RpYN06 in the *ORF1ab* tree (BP = 100) and linked to RaTG13 in the *Spike* gene tree (BP = 100). Our phylogenetic CGB barcodes showed that these discordances are explained by the mosaic structure of the SARS-CoV-2 genome, in which different regions support different relationships. Although SARS-CoV-2 was closely related to RaTG13 based on four genomic regions, representing only 4.8% of the WG alignment, other genomic regions provided support for 51 other phylogenetic relationships, including its grouping with RpYN06 (two regions; 5.8%), RmYN02 (two regions; 3.8%), RpBANAL103 (one region; 3.3%), RshSTT200 (two regions; 2.3%), etc. (Figure 5).

Based on amino-acid sequences of the receptor-binding domain (RBD) to the cellular ACE2 receptor, Temmam et al. [15] published a tree showing a close relationship between SARS-CoV-2 and three *Rhinolophus* viruses detected in Laos, i.e., RmBANAL52, RpBANAL103, and RmaBANAL236. In our nucleotide whole-genome alignment, the RBD region corresponds to pos. 22,675–23,352, and phylogenetic CGB barcodes indeed confirmed that a robust phylogenetic signal exists to support the node uniting SARS-CoV-2 and the three viruses from Laos, but it is restricted to pos. 22,801–23,250 of RBD (“bipartition +3” n°18 in Figure 5). Interestingly, upstream and downstream regions support other relationships for SARS-CoV-2, including its grouping with RaTG13 and MjGuangxi (pos. 22,501–22,700) or with RshSTT200 (pos. 24,001–24,250). More generally, and focusing only on the best bipartition categories +1 to +5, other genomic regions of SARS-CoV-2 were found to be closely related to one (viruses with asterisk) or more of the following ten sarbecoviruses (20 bipartitions representing 43% of the robust phylogenetic signals in the WG alignment): RaTG13*, RmBANAL52*, RpBANAL103*, RmarBANAL236*, RmBANAL247*, RmYN02*, RpYN06*, RshSTT200*, MjGuangxi, and RacCS203. Although several of these bipartitions represent compatible nested bipartitions (for example, “bipartitions +1” n°1 and n°3 are nested within “bipartition +2” n°9 in Figure 5), most of them support incongruent phylogenetic relationships.

Phylogenetic CGB barcodes allowed us to detect that several gene fragments provide conflicting phylogenetic signals. The best example concerns the *Spike* gene (pos. 21,664–25,557 in our alignment) for which we found conflicting phylogenetic signals supporting the grouping of SARS-CoV-2 with RaTG13 (pos. 21,701–22,400), RshSTT200 (pos. 24,001–24,250 and 25,401–25,850), RmBANAL247 + RmYN02 (pos. 24,951–25,350), MjGuangxi + RaTG13 (pos. 22,501–22,700), RmBANAL52 + RmaBANAL236 + RpBANAL103 (pos. 22,801–23,250), and RaTG13 + RmBANAL52 + RmaBANAL236 + RpBANAL103 + RshSTT200 (pos. 23,351–23,800) (Figure 5). Similarly, we found conflicting phylogenetic signals supporting the grouping of SARS-CoV with RaLYRa11 (pos. 22,801–23,100), Rs7327 + Rs9401 (pos. 24,151–25,400), and Rs4231 + Rs4874 (pos. 21,401–22,700) (Figure 3). We conclude, therefore, that phylogenetic CGB barcodes provide much more reliable and accurate information than WG and gene trees for understanding the evolution of sarbecoviruses.

### 4.2. Intermediary Hosts for SARS-CoV and SARS-CoV-2?

While many studies agree that bats are the main reservoir host for sarbecoviruses, the role of other mammals as possible intermediate hosts between bats and humans, such as small carnivores and pangolins, remains unclear and controversial [4,5,6,7,9,14,15,22]. To better understand this issue, we have included in our analyses several sarbecoviruses sequenced from captive mammals, including pangolins, minks and civets (Appendix A).

In theory, pangolins could be contaminated in their natural habitat by pathogens circulating in horseshoe bats because both taxa are occasionally found together in hollow trees, burrows and possibly caves [14,22]. In contrast to bats, which are considered as asymptomatic for coronavirus, pangolins were found to be highly sensitive to sarbecovirus [23]. In addition, pangolins are not gregarious like *Rhinolophus* bats; they are solitary species and the female and male meet only for reproduction. Therefore, most pangolins infected by bat sarbecovirus in the wild should be considered as evolutionary dead ends for the virus. However, the situation changed significantly with the intense pangolin trafficking during the decade before COVID-19, as Sunda pangolins (*M. javanica*) imported illegally into China became infected in captivity [14]. In this way, at least two different pangolin sarbecoviruses were exported from Southeast Asia to China, MjGuangxi before 2017 and MjGuangdong before 2019 [14,22,23,24]. Despite this, we did not find any evidence of exclusive ancestry between SARS-CoV-2 and the two pangolin viruses. Moreover, the synonymous nucleotide compositions of MjGuangxi and MjGuangdong genomes were found to be similar but divergent from those of SARS-CoV-2 and bat *SCoV2rC* viruses, suggesting that the two pangolin viruses have evolved independently and for some time in pangolin populations in the wild or in captivity [16]. Although pangolins may be intermediary hosts between bats and humans for some viruses, current data and our findings do not support their involvement in the case of SARS-CoV-2.

Recent studies have provided strong evidence that SARS-CoV-2 was introduced from humans to domestic or captive carnivores, including dogs, cats, lions, tigers and minks, and that the virus evolved very rapidly in mink farms, with several back transmissions from infected animals to humans [25,26]. The common ancestor of human and mink SARS-CoV-2 genomes was supported by high MBP values (82/99), confirming that a strong phylogenetic signal is present in all parts of the alignment (as highlighted by the green colour of GB_400_ and GB_2000_ barcodes in Figure 2). In addition, we found no genomic region providing support for the paraphyly or polyphyly of SARS-CoV-2. These results therefore confirmed that human and mink SARS-CoV-2 genomes included in our study share the same MRCA, fully isolated genetically from bat *Sarbecovirus* lineages since its emergence in December 2019 or a few weeks earlier. Our analyses also revealed that different genomic regions can support conflicting relationships between SARS-CoV-2 sequences. The best example concerns the placement of Gamma and Omicron variants, which were related to the Delta variant based on pos. 9851–10,350 or to Alpha variant based on pos. 28,801–29,350. These discordant patterns could be due to past genomic recombination, as several recombinants have already been reported in human populations [11,12].

The masked palm civet (*Paguma larvata*) was identified as the possible intermediate host transmitting SARS-CoV to humans during the SARS epidemic, which began on November 2002 in Foshan, a city about 20 km from Guangzhou, the largest city of Guangdong province (China) [13,27]. In our analyses, we have included three SARS-CoV-like genomes detected in civets maintained in captivity in the Guangdong province: two came from a wildlife market in Shenzhen before May 2003 (PlSZ3 and PlSZ61) and another was sampled in a restaurant in Guangzhou in 2003 (PlGZ81). In the WG tree of Figure 2, it is worth noting that human and civet SARS-CoVs are enclosed into a robust clade (BP_WG_ = 100) also supported by high MBP values (74/100), which indicates that the phylogenetic signal is strong in all parts of the whole-genome alignment. These results therefore confirmed that human and civet SARS-CoV genomes share the same MRCA, which was fully isolated genetically from bat *Sarbecovirus* lineages. Although the node grouping the four human SARS-CoVs was not robust in the WG tree (BP_WG_ = 52), it was supported by two GRPS, one including the RBD region (pos. 22,651–24,200 in our alignment) and another covering the C-terminus of the Spike protein and the N-terminus of ORF3a (pos. 25,251–25,750). Interestingly, there are two RBD amino-acids characterizing the four human SARS-CoVs included in our alignment, F360 and T487. The last one was found to increase by 20-fold the RBD affinity for human ACE2, suggesting therefore a specific adaptation to enhance human-to-human transmission during the 2002–2003 SARS-CoV outbreak [28]. In contrast, all human and animal viruses sequenced during the 2003–2004 SARS-CoV outbreak had a Serine (instead of Threonine) at this position of the Spike protein [13,28], indicating that this new episode may have resulted from an independent viral invasion from animal to human [29]. Our analyses have shown that several discordant phylogenetic signals involving the para- or polyphyly of human SARS-CoVs were detected in other genomic regions, such as the grouping of the three civet SARS-CoVs with HsRef and HsShanghaiQXC1 or the grouping of PlSZ3 with HsRef, HsShanghaiQXC1 and HsGD01. Therefore, these results suggest that human and civet SARS-CoVs were involved in past genomic recombination events in the same host and that humans and captive civets have exchanged SARS-CoV viruses. In other words, the situation was probably very similar to that recently observed for SARS-CoV-2 between humans and captive minks [25,26]. Due to the low divergence separating human and civet genomes (between 0.16% and 0.41%), however, we cannot rule out two other hypotheses involving either nucleotide homoplasy (due to convergence(s) and reversion(s)) or sequencing and genome assembly errors.

### 4.3. Species Involved as Reservoir Hosts for the Ancestors of SARS-CoV and SARS-CoV-2

As discussed previously [9,10], most conflicting phylogenetic signals in different genomic regions of our *Sarbecovirus* alignment can be explained by multiple past events of recombination at different periods of time and involving viruses circulating in several reservoir species of horseshoe bats (genus *Rhinolophus*). Host CGB barcodes obtained for both SARS-CoV and SARS-CoV-2 (Figure 3 and Figure 5) clearly confirm that horseshoe bats are the animal reservoir in which coronaviruses related to either SARS-CoV (*SCoVrC*) or SARS-CoV-2 (*SCoV2rC*) evolve and diversify. First, all animal sarbecoviruses showing exclusive ancestry with human sarbecoviruses were extracted from *Rhinolophus* species. Second, 100% of the phylogenetic CGB barcodes reconstructed for either SARS-CoV or SARS-CoV-2 involved *Rhinolophus* viruses. The *Rhinolophus* species showing the highest contribution are *R. ferrumequinum* (76%), *R. sinicus* (71%), and *R. affinis* (58%) for SARS-CoV, and *R. pusillus* (82%), *R. malayanus* (81%), *R. marshalli* (65%) and *R. affinis* (60%) for SARS-CoV-2 (Figure 4B and Figure 6B). All these results corroborate high and recent evolutionary dynamics of genomic recombination between sarbecoviruses circulating in several *Rhinolophus* species. As pointed out in previous publications [9,16,22], recombination implies that whole or partial genomes of two divergent viruses co-exist in a cell of the same host. Such a situation is expected to occur frequently in horseshoe bats because interspecific transmission of sarbecoviruses could be favoured by their behaviour, as several *Rhinolophus* species often nest in colonies in the same cave, and by their cave habitat, in which viral contamination could be facilitated by promiscuity within and between bat colonies, high humidity levels and cool temperatures all year round.

The ecological niches (i.e., geographic distributions predicted using climatic parameters) of bat sarbecoviruses related to either SARS-CoV (*SCoVrC*) or SARS-CoV-2 (*SCoV2rC*) have been reconstructed using an original approach combining genetic data on both viruses and bat species [22]. The results have shown that the ecological niche of *SCoVrC* extends from Southwest China and northern Myanmar, through northern Vietnam and Central China to East China, Korea and southern Japan, whereas the ecological niche of *SCoV2rC* includes four main different regions of Southeast Asia: (i) northern Laos and bordering regions; (ii) southern Laos, southwestern Vietnam, and north-eastern Cambodia; (iii) the East region of Thailand and southwestern Cambodia; and (iv) the Dawna Range in central Thailand and south-eastern Myanmar. Since the geographic ranges of *SCoVrC* and *SCoV2rC* are different, we assumed that these two groups of viruses generally do not circulate in the same *Rhinolophus* species assemblages. In agreement with that, geographic CGB barcodes indicate that 84% of the SARS-CoV genome show exclusive ancestry with bat viruses from Southwest China (Figure 4C), whereas 81% of the SARS-CoV-2 genome show exclusive ancestry with bat viruses from Southeast Asia and Yunnan (Figure 6C). In addition, host CGB barcodes show that SARS-CoV shares exclusive ancestry with several viruses found in three bat species mainly distributed in China [30]: 19.4% for *R. ferrumequinum*, 13.1% for *R. sinicus*, and 1.3% for *R. pearsonii* (Figure 4B), whereas SARS-CoV-2 shares exclusive ancestry with several viruses found in three bat species mainly distributed in Southeast Asia or endemic to this region [30]: 6.8% for *R. malayanus*, 2.3% for *R. shameli*, and 1.7% for *R. marshalli* (Figure 6B).

However, the detection of recombinant viruses between *SCoVrC* and *SCoV2rC* lineages, named *RecSar* [9,16], has revealed that some bats were simultaneously infected by the two divergent virus lineages in the past. Interestingly, different viruses showing exclusive ancestry with either SARS-CoV or SARS-CoV-2 were isolated from *R. pusillus* (representing 3.3% and 9.1%, respectively) and also from *R. affinis* (representing 1.0% and 4.8%, respectively). Since these two *Rhinolophus* species are widely distributed in both China and Southeast Asia [30], their dispersal capacity is expected to be greater than that of other *Rhinolophus* species endemic to either China or Southeast Asia. Therefore, it can be argued that the rare events of genomic recombination between *SCoVrC* and *SCoV2rC* lineages may have occurred in *R. affinis* and *R. pusillus* bats and most likely in the region where the ecological niches of *SCoVrC* and *SCoV2rC* overlap, i.e., southern Yunnan and adjacent regions in northern Laos, and border regions between Laos and Vietnam [22].

### 4.4. Conclusion and Perspectives

Exclusive ancestry with human sarbecoviruses has currently been found in sarbecoviruses collected in the following *Rhinolophus* species (Figure 3 and Figure 5): *R. ferrumequinum*, *R. sinicus*, and *R. pearsonii* for SARS-CoV; *R. malayanus*, *R. shameli*, and *R. marshalli* for SARS-CoV-2; and *R. pusillus* and *R. affinis* for both SARS-CoV and SARS-CoV-2. These results provide strong evidence that viral transmission within and between bat colonies of these species as well as genomic recombination have participated to the emergence of human sarbecoviruses. However, the lists of involved reservoir species cannot be considered exhaustive due to limited investigations for detecting sarbecoviruses in bats, particularly in Southeast Asia where the diversity of *Rhinolophus* species is much higher than anywhere else in the Old World [30]. Although genome data currently available support an origin of SARS-CoV in horseshoe bats of Yunnan and an origin of SARS-CoV-2 in horseshoe bats of Yunnan and northern Laos, the two hypotheses need to be further investigated by exploring *Sarbecovirus* diversity in bats of Cambodia, Laos, Myanmar, Thailand and Vietnam. Indeed, only a few bat sarbecoviruses were recently found in a limited number of localities in northern Cambodia (two highly similar genomes of the same virus collected in one locality) [5], eastern Thailand (five highly similar genomes of the same virus collected in one locality) [6] and northern Laos (five viruses; four localities) [15], and no sarbecovirus has yet been described from bats of Myanmar and Vietnam.

Another problem that limits our interpretation on reservoir hosts is that most studies dealing with the evolution of sarbecoviruses have generally provided very little information about the bats and caves in which the viruses were found. Recently, a new *Sarbecovirus* genome, RspSC2018 (GenBank accession: MK211374), was described from an unidentified bat at the species level collected from an unknown locality in Sichuan province [31]. Since Illumina reads were not deposited in any of the international Sequence Read Archive (SRA) databases, it was impossible to know from which host species the virus was extracted. Moreover, the quality of the virus genome assembly cannot be verified, which may also compromise some future evolutionary studies.

Our poor knowledge of horseshoe bat taxonomy also hinders understanding of their role as *Sarbecovirus* reservoir. For instance, some taxonomists have proposed to split *R. ferrumequinum* into two different species: *R. ferrumequinum* in Europe and West Asia, and *Rhinolophus nippon* in East Asia [32]. Similarly, a molecular study based on a set of ~1500 nuclear loci has suggested that the *R. sinicus* complex may contain three different species [33]: *R. sinicus*, apparently distributed from northern Vietnam to East China, through Hainan Island and Central China; *Rhinolophus septentrionalis* in Yunnan; and an undescribed species of *Rhinolophus* in Vietnam.

## Figures and Tables

**Figure 1 viruses-15-00406-f001:**
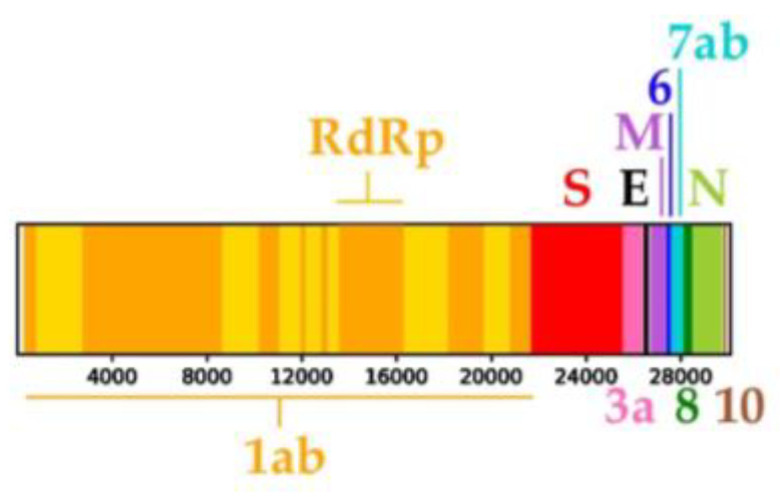
Positions of the coding sequences in the alignment of 75 *Sarbecovirus* genomes (30,115 nucleotides). For convenience, the size and scale are the same as the coloured genomic bootstrap (CGB) barcodes shown in Section 3.2 and Section 3.3. Abbreviations: *E*: envelope gene; *M*: membrane gene; *N*: nucleocapsid gene; *RdRp*: RNA-dependent RNA polymerase gene; *S*: spike gene; 1ab: ORF (Open Reading Frame) 1ab; 3a: ORF3a; 6: ORF6; 7ab: ORF7ab; 8: ORF8; 10: ORF10. The alternating yellow and orange colours in ORF1ab indicate different non-structural proteins, including *RdRp*.

**Figure 2 viruses-15-00406-f002:**
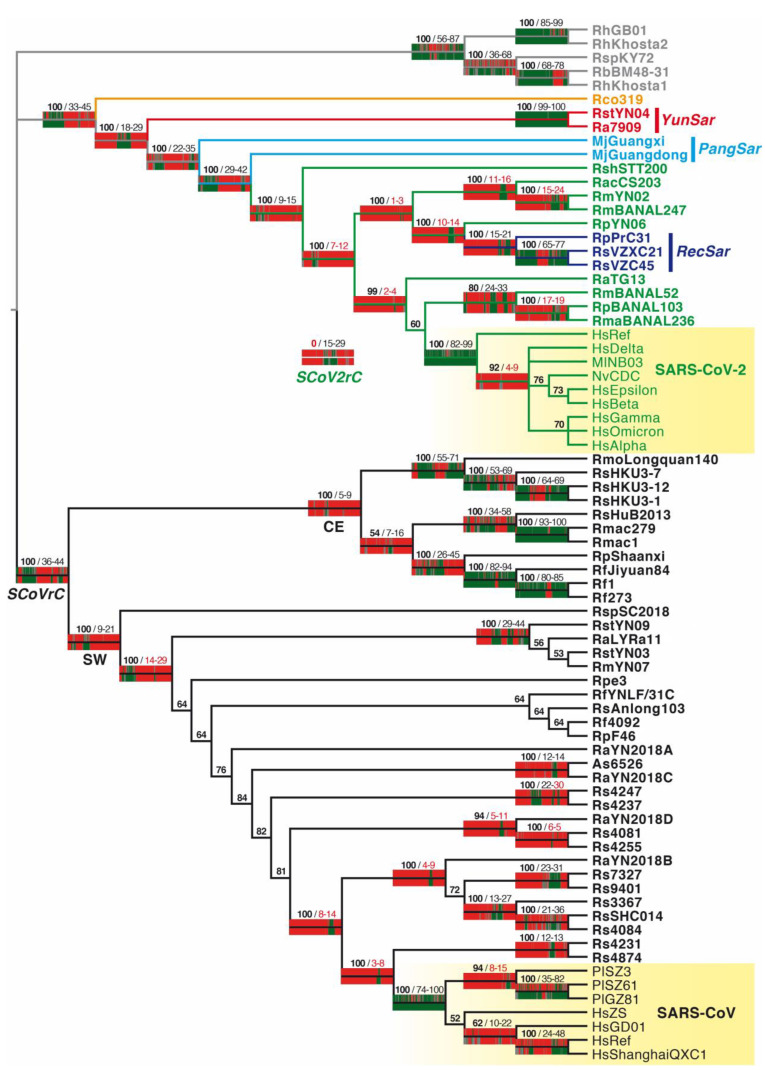
Whole-genome bootstrap (WGB) tree reconstructed from the alignment of 75 *Sarbecovirus* genomes. This is a 50% majority-rule consensus reconstructed after 1000 bootstrap replicates using the RAxML method and a GTR+G model for each of the four partitions of the whole-genome alignment (three codon positions and non-coding regions). The bootstrap percentages (BP_WG_) are given for all nodes in bold. The genomic bootstrap (GB) barcodes built from SWB_400_ and SWB_2000_ outputs and the BBC program [9] were reported for all nodes recovered as monophyletic in the SuperTRI bootstrap (SB) trees reconstructed from SWB analyses (for these nodes, MBP_400_ and MBP_2000_ values are indicated in black at the right of the slash) and for all nodes supported by BP_WG_ ≥ 90% that were not found as monophyletic in SB trees (for these nodes, MBP_400_ and MBP_2000_ values are indicated in red at the right of the slash). The GB_400_ and GB_2000_ barcodes are displayed above and below the branches, respectively. The colours of Asian sarbecoviruses indicate to which group of synonymous nucleotide composition they belong [16]: black for SARS-CoV related coronaviruses (*SCoVrC*), green for SARS-CoV-2 related coronaviruses (*SCoV2rC*), dark blue for *RecSar* viruses showing evidence of genomic recombination between *SCoVrC* and *SCoV2rC*, light blue for pangolin sarbecoviruses, red for *YunSar* viruses and orange for the Rco319 virus from Japan. For comparison, the GB_400_ and GB_2000_ barcodes supporting the monophyly of *SCoV2rC* are shown in the middle of the figure.

**Figure 3 viruses-15-00406-f003:**
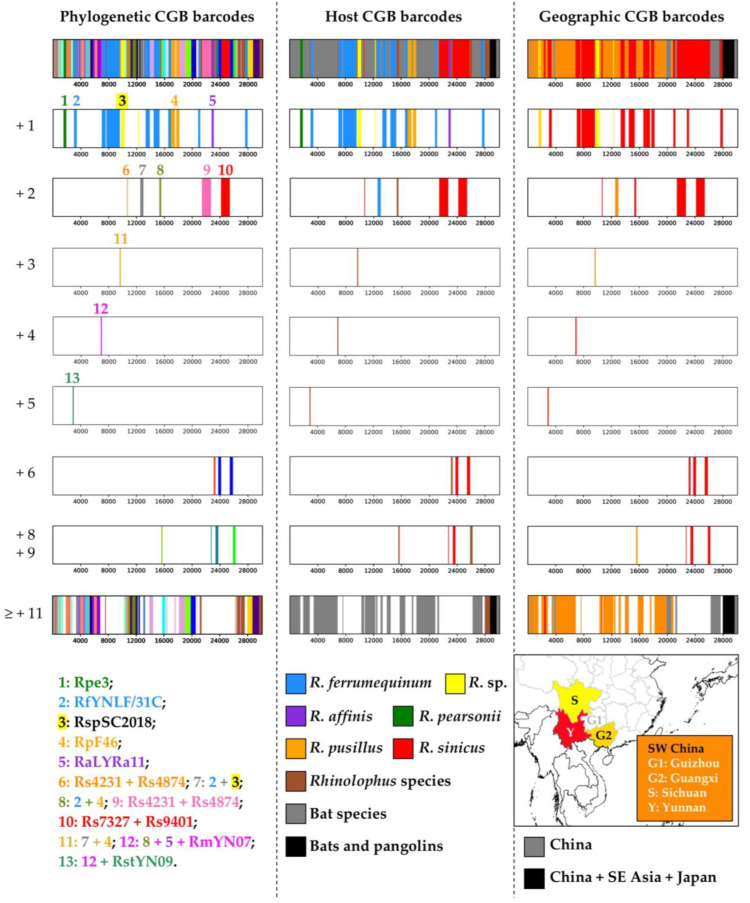
Coloured genomic bootstrap (CGB) barcodes constructed for the common ancestor of SARS-CoV. At the left part of the figure are shown phylogenetic CGB barcodes, in which the best phylogenetic signals (i.e., BBC bipartitions including the fewest number of closely-related viruses) are represented by different colours. To facilitate interpretation, we have also shown versions reduced to the bipartition categories +1 (n = 5 bipartitions including SARS-CoV and another virus), +2 (n = 5 bipartitions including SARS-CoV and two other viruses), +3 (n = 1), +4 (n = 1), +5 (n =1), +6 (n = 2), +8 and +9 (n = 4) and all bipartitions uniting SARS-CoV sequences with at least 11 other viruses (n = 30). The bat sarbecoviruses included in the 13 smallest bipartitions (categories +1 to +5) are detailed at the bottom. Similarly, the full and reduced versions of host and geographic CGB barcodes are shown in the middle and right parts of the figure. The colour codes used for host taxa and geographic areas are provided at the bottom.

**Figure 4 viruses-15-00406-f004:**
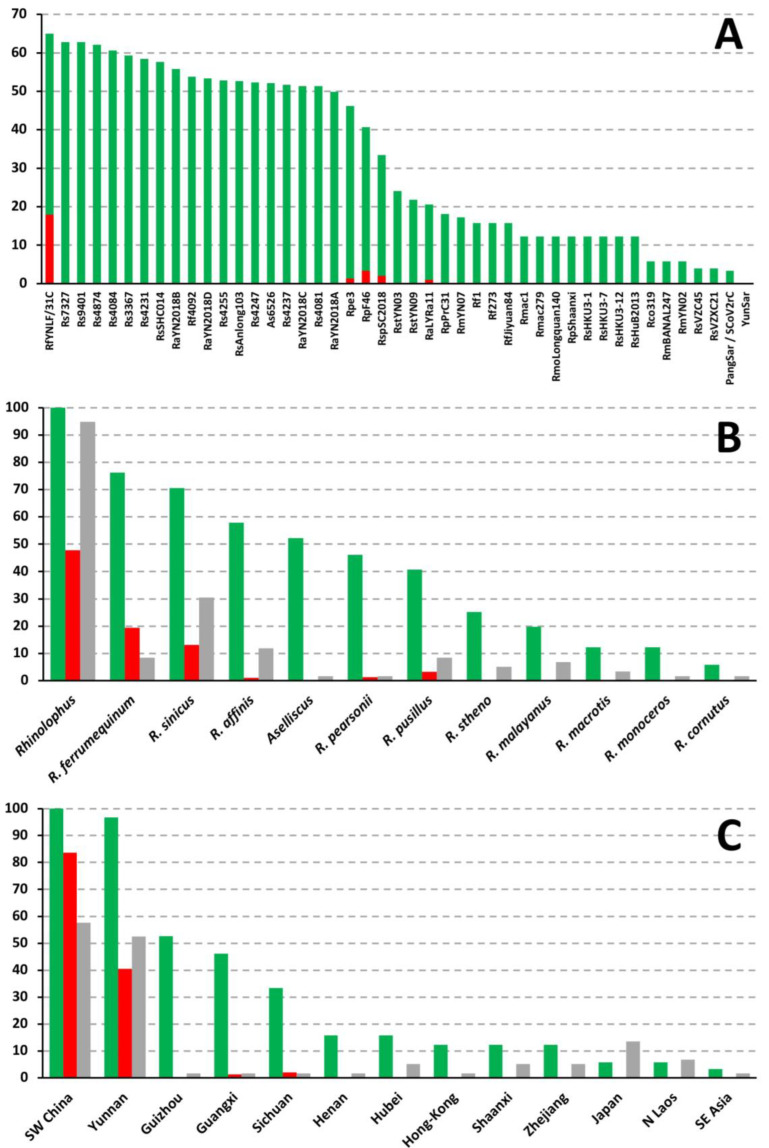
Percentages of whole-genome alignment including phylogenetic CGB barcodes (green histograms) shared between SARS-CoV and several bat sarbecoviruses (**A**) detected in different host taxa (**B**, green histograms) and geographic regions (**C**, green histograms). The red histograms indicate the percentages of exclusive ancestry found in bat sarbecoviruses (**A**), host taxa (**B**) and geographic regions (**C**). To assess sampling efforts, the grey histograms show the proportions of bat viruses used in our dataset for the host taxa and geographic regions of interest, respectively.

**Figure 5 viruses-15-00406-f005:**
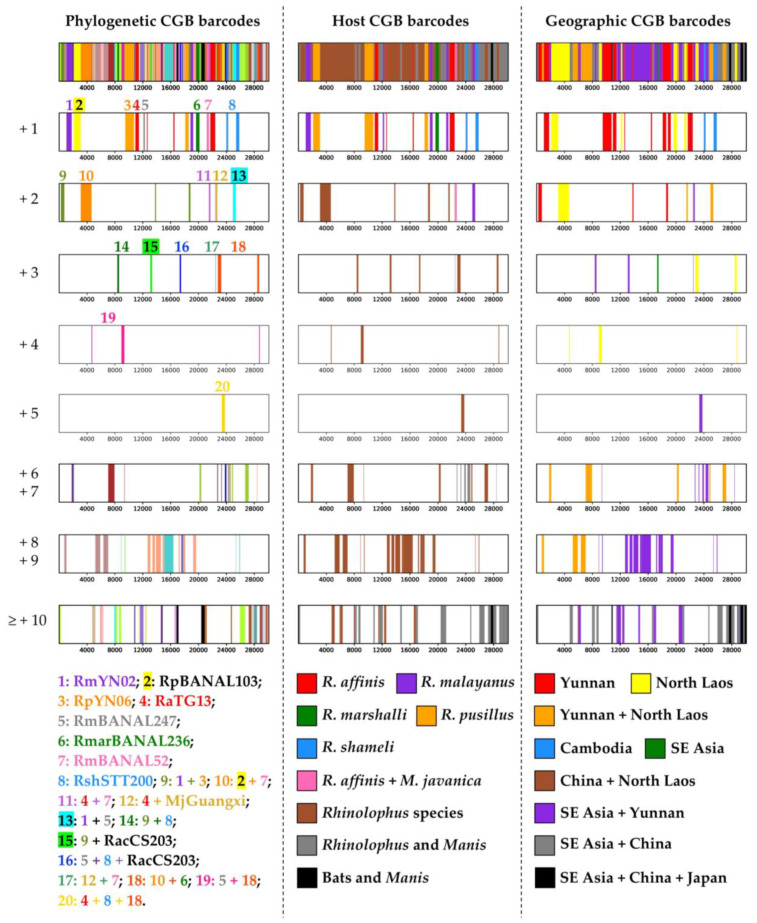
Coloured genomic bootstrap (CGB) barcodes constructed for the common ancestor of SARS-CoV-2. At the left part of the figure are shown phylogenetic CGB barcodes, in which the best phylogenetic signals (i.e., BBC bipartitions including the fewest number of closely-related viruses) are represented by different colours. To facilitate interpretation, we have also shown versions reduced to the bipartition categories +1 (n = 8 bipartitions including SARS-CoV-2 and another virus), +2 (n = 5 bipartitions including SARS-CoV-2 and two other viruses), +3 (n = 5), +4 (n = 1), +5 (n = 1), +6 and +7 (n = 8), +8 and +9 (n = 6), and all bipartitions uniting SARS-CoV-2 sequences with at least 10 other viruses (n = 18). The bat and pangolin sarbecoviruses included in the 20 smallest bipartitions (categories +1 to +5) are detailed at the bottom. Similarly, the full and reduced versions of host and geographic CGB barcodes are shown in the middle and right parts of the figure. The colour codes used for host taxa and geographic areas are provided at the bottom.

**Figure 6 viruses-15-00406-f006:**
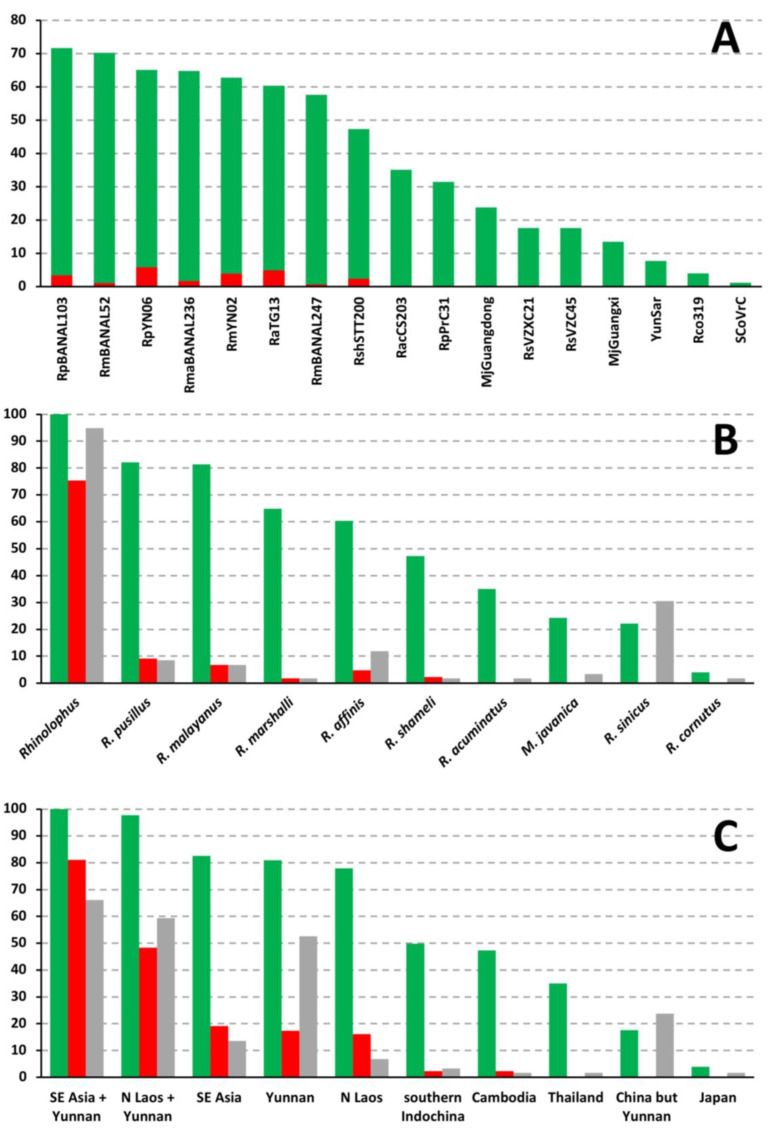
Percentages of whole-genome alignment including phylogenetic CGB barcodes (green histograms) shared between SARS-CoV-2 and several bat or pangolin viruses (**A**) detected in different host taxa (**B**, green histograms) and geographic regions (**C**, green histograms). The red histograms indicate the percentages of exclusive ancestry found in bat sarbecoviruses (**A**), host taxa (**B**) and geographic regions (**C**). To assess sampling efforts, the grey histograms show the proportions of sarbecoviruses used in our dataset for the host taxa and geographic regions of interest, respectively.

**Table 1 viruses-15-00406-t001:** Five sliding window bootstrap (SWB) analyses performed on an alignment of 75 *Sarbecovirus* genomes (length: 30,115 nt).

	Window Size (nt)	Step (nt)	WB Sub-datasets	SWB Bipartitions ^1^	SuperTRI Matrix ^2^	BBC Bipartitions ^3^	SARS-CoV Bipartitions ^4^	SARS-CoV-2 Bipartitions ^4^
**1**	400	50	595	204,001	750,330	1263	215	178
**2**	500	50	593	150,118	649,437	1240	208	176
**3**	600	50	591	116,934	577,232	1212	213	175
**4**	1000	50	583	57,444	427,041	1061	177	163
**5**	2000	50	563	19,875	288,711	818	142	132

^1^: Number of bipartitions (with window bootstrap percentages (WBP) calculated for each WB subdatasets) obtained under the SWB program [9]; ^2^: Number of characters in the MRP matrix reconstructed using LFG [9] and SuperTRI [21] programs; ^3^: Number of SWB bipartitions including at least one WBP ≥ 50% selected under the BBC program using the SWB file as input [9]; ^4^: Number of BBC bipartitions selected under Microsoft^®^ Excel using the BBC file as input.

## Data Availability

The whole-genome alignment, the five BBC output files, the five SuperTRI MRP files, the five SB trees (SB_400_, SB_500_, SB_600_, SB_1000_, and SB_2000_), all the 5594 GB barcodes, the two Excel and six CSV files used to construct CGB barcodes, and all the 303 CGB barcodes constructed in this study are available in the Open Science Framework (OSF) platform at https://osf.io/nj57e/. The SWB, BBC, CGB and LFG programs are available at https://github.com/OpaleRambaud/GB_barcodes_project.

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
