# Peer review of "Retracing Phylogenetic, Host and Geographic Origins of Coronaviruses with Coloured Genomic Bootstrap Barcodes: SARS-CoV and SARS-CoV-2 as Case Studies"

_viruses, 2023, doi:10.3390/v15020406_

Round 1
Reviewer 1 Report
As one of the highlights of this article, the new method CGB barcodes may be described in more details in the Method section.
It seems to lack a clear & explicit definition of GRPS.
The process of constructing phylogenetic, host and geographic CGB barcodes from SWB bipartitions is unclear.
The structure of CGB barcodes (especially for host and geographic) and their display in fig 3 & 5 need a clear explanation.
In fig 3 & 5, the defination of bipartition categories is unclear.
Author Response
Reviewer#1
As one of the highlights of this article, the new method CGB barcodes may be described in more details in the Method section.
It seems to lack a clear & explicit definition of GRPS.
The criteria used to identify the GRPS (genomic regions containing robust phylogenetic signal) are already fully described in section 2.4 (see paragraph below). Considering that Reviewer #1 may have misinterpreted something, the paragraph has been changed slightly as highlighted in red below:
“In the third step, all WBPs ≥ 70% were highlighted in green and all WBPs comprised between 50% and 70% were highlighted in yellow green using conditional formatting options in Microsoft® Excel. We performed the comparisons starting with bipartitions +1, i.e. containing all SARS-CoV-2 sequences plus one and another additional virus. Due to past events of genomic recombination, we found several bipartitions +1 supporting conflicting phylo-genetic relationships. For each of these BBC bipartitions, we identified the intervals of genomic regions containing robust phylogenetic signal (GRPS) using the criteria developed later below. Then we proceeded similarly by analysing other genomic fragments for bipartitions +2 (containing all SARS-CoV-2 sequences plus and two additional other viruses), bipartitions +3 (containing all SARS-CoV-2 sequences plus and three additional other viruses), etc. By this way, we were able to identify the closest virus(es) to SARS-CoV-2 in all regions of our genome alignment. All selected GRPS contained a robust phylogenetic signal (WBP ≥ 70%) in at least two WB subdatasets of the BBC400, BBC500 or BBC600 results. The 5’ and 3’ ends of GRPS were extended using the following criteria: (i) by accepting WBPs between 50% and 70% for BBC400 results; (ii) when median positions showed an average WBP ≥ 50% for BBC400, BBC500 and BBC600 results; and (iii) when median positions showed an average WBP ≥ 50% for all the five BBC results (BBC400, BBC500, BBC600, BBC1000, and BBC2000). This strategy was adopted for three major reasons: (i) GRPS of small lengths cannot be detected using SWB analyses based on the largest window sizes (i.e. 1000 nt and 2000 nt); (ii) due to stochastic variation in BP bootstrap values, the comparisons between the three SWB analyses based on the smallest window sizes (i.e. 400 nt, 500 nt and 600 nt) allow us to better detect GRPS of small lengths; (iii) GRPS of large size can be erroneously interrupted if we consider only SWB analyses based on the smallest window sizes because they contain lesser amounts of phylogenetic signal [9]. It is therefore important to also make comparisons with BBC results based on the largest window sizes (i.e. 1000 nt and 2000 nt).”
The process of constructing phylogenetic, host and geographic CGB barcodes from SWB bipartitions is unclear.
We tried to improve the process description as shown below:
“In the fourth step, the intervals of GRPS (5’ and 3’ median positions in the whole-genome alignment) were written in a new CSV file for each of the BBC bipartitions including SARS-CoV2 sequences in which one or more GRPS were identified as the best phylogenetic signals (i.e. containing the fewest number of closely-related viruses). A specific colour code was chosen for each of the selected SWB bipartitions, and the file was used as input in the CGB program (python script) to construct phylogenetic CGB barcodes of different colours. Two other files were derived from the original CSV file: (i) a file for host CGB barcodes, in which different colours were assigned to the BBC bipartitions including viruses extracted from the following nine taxa: Rhinolophus affinis, Rhinolophus malayanus, Rhinolophus marshalli, Rhinolophus pusillus, Rhinolophus shameli, R. affinis + Manis javanica; Rhinolophus species, Rhinolophus species + M. javanica, bat species + M. javanica; and (ii) a file for geographic CGB barcodes, in which different colours were chosen for the BBC bipartitions including viruses collected in the following nine geo-graphic areas: Cambodia; North Laos, SE Asia, Yunnan, North Laos + Yunnan, SE Asia + Yunnan, North Laos + China, SE Asia + China, and SE Asia + China + Japan. All files used to construct the CGB barcodes for SARS-CoV-2 and SARS-CoV (including BBC400, BBC500, BBC600, BBC1000, BBC2000 files, the two Excel files used to identify GRPS intervals, and the six CSV files used for phylogenetic, host and geographic CGB barcodes) are available at https://osf.io/nj57e/.
The structure of CGB barcodes (especially for host and geographic) and their display in fig 3 & 5 need a clear explanation.
In fig 3 & 5, the defination of bipartition categories is unclear.
The caption of Figure 3 and Figure 5 were modified as follows:
“Figure 3. Coloured genomic bootstrap (CGB) barcodes constructed for the common ancestor of SARS-CoV. At the left part of the figure are shown phylogenetic CGB barcodes, in which the best phylogenetic signals (i.e. bipartitions including the fewest number of closely-related viruses) are represented by different colours. To facilitate interpretation, we have also shown versions reduced to the bipartition categories +1 (n = 5 bipartitions including SARS-CoV and another virus), +2 (n = 5 bipartitions including SARS-CoV and two other viruses), +3 (n = 1), +4 (n = 1), +5 (n =1), +6 (n = 2), +8 and +9 (n = 4), and all bipartitions uniting SARS-CoV sequences with at least 11 other viruses (n = 30). The bat sarbecoviruses included in the 13 smallest bipartitions (categories +1 to +5) are detailed at the bottom. Similarly, the full and reduced versions of host and geographic CGB barcodes are shown in the middle and right parts of the figure. The colour codes used for host taxa and geographic areas are provided at the bottom.”
Reviewer 2 Report
I read with keen interest this paper which has successfully expanded on the author's previous work by investigating Sarbecoviruses phylogenetic, host and geographic origins. The unique selling point of this paper is the use of coloured genomic bootstraps (CGB) to visualise genomic regions according to their origins, revolutionising the phylogeographic field and providing a useful tool to investigate the evolution of viruses, particularly those prone to recombination. The results have been well situated within the current literature and indeed offer explanations to irregularities highlighted by previous publications, proving its value to the scientific community. The figures are easy to understand and help the reader navigate the complexities of the analysis. The methods are complex at times to understand, but once the open science framework link is made available this should help. The conclusions are sound, with most limitations being explored. Overall I think this manuscript is worthy of publication and furthers our understanding of SARS-CoV evolution.
I have just a few comments / queries listed below, but on the whole this is one of the easiest manuscripts I have reviewed recently in terms of quality of writing, so well done!
1. What measures did you put into place to minimise errors during the data manipulation in excel? Do you envisage future developments to remove this labour intensive process?
2. Please enlarge figure 1 to fit a page width so that its easier to see the genome regions.
3. 3. Have you investigated whether there are certain ‘hot spots’ for recombination across sarbecoviruses?
4. Although in the conclusions section you discuss the limitations of unsampled viruses, I also think its important to mention this in the discussion section.
a. For example in section 4.2 (line 585-587) when discussion pangolin involvement, I think a sentence to recognise that unsampled viruses may exist (or have existed) to change your findings.
b. In paragraph 663-683, sampling biases may affect your analysis.
5. Also in paragraph 663-683, can you comment on whether the Rhinolophus species you are discussing travel long distances, which will enhance the possibility of viruses mixing.
6. Please add a reference for sentence 656-661 'Such a situation.....all year round'.
7. In the discussion please can you a sentence or two on the pros and cons of using this method in comparison to other more established methods, and also comment on the possibility of including temporal data (sampling date) into future improvements as other analysis applications such as BEAST.
Author Response
Reviewer#2
I read with keen interest this paper which has successfully expanded on the author's previous work by investigating Sarbecoviruses phylogenetic, host and geographic origins. The unique selling point of this paper is the use of coloured genomic bootstraps (CGB) to visualise genomic regions according to their origins, revolutionising the phylogeographic field and providing a useful tool to investigate the evolution of viruses, particularly those prone to recombination. The results have been well situated within the current literature and indeed offer explanations to irregularities highlighted by previous publications, proving its value to the scientific community. The figures are easy to understand and help the reader navigate the complexities of the analysis. The methods are complex at times to understand, but once the open science framework link is made available this should help. The conclusions are sound, with most limitations being explored. Overall I think this manuscript is worthy of publication and furthers our understanding of SARS-CoV evolution.
I have just a few comments / queries listed below, but on the whole this is one of the easiest manuscripts I have reviewed recently in terms of quality of writing, so well done!
We would like to acknowledge reviewer#2 for her/his nice comments on our manuscript.
- What measures did you put into place to minimise errors during the data manipulation in excel? Do you envisage future developments to remove this labour intensive process?
We fully agree with reviewer #2 that this labour intensive process needs to be automated. However, this seems for now quite difficult in terms of programming. We are aware that our new method is tedious and may appear time-consuming to implement. However, we consider that the results are very interesting for interpreting the evolution of recombining viruses. To help the readers, all files generated during this study will be available at https://osf.io/nj57e/, including the whole-genome alignment, the five BBC output files, the five SuperTRI MRP files, the five SB trees (SB400, SB500, SB600, SB1000, and SB2000), all the 5,594 GB barcodes, the two Excel files and six CSV files used to construct CGB barcodes, and all the 303 CGB barcodes constructed in this study.
- Please enlarge figure 1 to fit a page width so that its easier to see the genome regions.
This first figure was drawn to allow direct comparisons with CGB barcodes shown in Figures 3 and 5. For that reason, we would like to keep its size unchanged. To be more precise, we have modified the caption as follows:
“Positions of the coding sequences in the alignment of 75 Sarbecovirus genomes (30,115 nucleotides). For convenience, the size and scale is are the same as the coloured genomic bootstrap (CGB) barcodes shown in Figures 3 and 5. Abbreviations: E: envelope gene; M: membrane gene; N: nucleocapsid gene; RdRp: RNA-dependent RNA polymerase gene; S: spike gene; 1ab: ORF (Open Reading Frame) 1ab; 3a: ORF3a; 6: ORF6; 7ab: ORF7ab; 8: ORF8; 10: ORF10. The alternating yellow and orange colours in ORF1ab indicate different non-structural proteins, including RdRp.”
Please note that our whole-genome alignment will be available at https://osf.io/nj57e/ and the intervals of the different genes are detailed in the main text as follows:
“In this study, 75 genomes of the subgenus Sarbecovirus (supplementary Table S1) were aligned to infer phylogenetic relationships. The positions of the coding sequences were the following in our final alignment of 30,115 nucleotides (nt): 256-21,654 for ORF1ab, including the RNA-dependent RNA polymerase gene [RdRp] at positions 13,540-16,335; 21,664-25,557 for the spike (S) gene; 25,567-26,394 for ORF3a; 26,419-26,649 for the envelope (E) gene; 26,704-27,375 for the membrane (M) gene; 27,388-27,579 for ORF6; 27,589-28,101 for ORF7ab; 28,108-28,485 for ORF8; 28,504-29,775 for the nucle-ocapsid (N) gene; and 29,803-29,919 for ORF10 (Figure 1).”
- 3. Have you investigated whether there are certain ‘hot spots’ for recombination across sarbecoviruses?
We acknowledge reviewer#2 for this interesting issue. However, to study this question, it would be necessary to analyze at least thirty different sarbecoviruses as we did for SARS-CoV and SARS-CoV-2 which represents an enormous amount of work.
- Although in the conclusions section you discuss the limitations of unsampled viruses, I also think its important to mention this in the discussion section.
- For example in section 4.2 (line 585-587) when discussion pangolin involvement, I think a sentence to recognise that unsampled viruses may exist (or have existed) to change your findings.
As indicated in section 4.2, “we did not find any evidence of exclusive ancestry between SARS-CoV-2 and the two pangolin viruses. Moreover, the synonymous nucleotide compositions of MjGuangxi and MjGuangdong genomes were found to be similar but divergent from those of SARS-CoV-2 and bat SCoV2rC viruses [16], suggesting that the two pangolin viruses have evolved independently and for some time in pangolin populations in the wild or in captivity. Although pangolins may be intermediary hosts between bats and humans for some viruses, current data and our findings do not support their involvement in the case of SARS-CoV-2.”
More arguments on this topic are provided in the recent article published by Hassanin A. (2022). Variation in synonymous nucleotide composition among genomes of sarbecoviruses and consequences for the origin of COVID-19. Gene, 835, 146641. https://doi.org/10.1016/j.gene.2022.146641
- In paragraph 663-683, sampling biases may affect your analysis.
We previously discussed the impact of sampling biases in our articles on ecological niches.
We consider this issue somewhat off topic here.
- Also in paragraph 663-683, can you comment on whether the Rhinolophus species you are discussing travel long distances, which will enhance the possibility of viruses mixing.
Unfortunately, many aspects of Rhinolophus evolution are completely unknown. Although some data based on radio telemetry are available for European species, the dispersal capacity of Asian species still needs to be explored.
- Please add a reference for sentence 656-661 'Such a situation.....all year round'.
It is our own interpretation. Indeed, AH had extensive field experiences with bats in Southeast Asia during the last twenty years. For instance, he visited many caves in Cambodia, Laos and Vietnam where several species of horseshoe bats were found in sympatry.
- In the discussion please can you a sentence or two on the pros and cons of using this method in comparison to other more established methods, and also comment on the possibility of including temporal data (sampling date) into future improvements as other analysis applications such as BEAST.
To be honest, it seems very complicated to carry out molecular dating on viruses evolving with such levels of recombination. This involves developing new methods.
Reviewer 3 Report
1. The authors introduced new method CGB barcodes on SARS-CoV and SARS-CoV-2 . Is it first report of CGB method?
2. Did CGB method use in other viruses ? If there has reports on other viruses, please include it in the discussion part.
2. It is better to describe limitation of CGB method.
3. This study indicate host and geographic origins of SARS-CoV & SARS-CoV-2 (Sarbecovirus Lineage B). Why the authors does not includes other Sarbecovirus like as Bat SL-CoV-WIV1 RaTG13?
4. The author described that SARS-CoV-2 share ancestry with Rinolophus viruses from Yunnan, Lao and Cambodia. How about relation with other countries?
5. In this study, the authors described genome, host taxa, geographic regions sharing between SARS-CoV-2 and several bats or pangolin viruses within Asian countries. Kindly explain the reason why you focused on Asia countries.
Author Response
Reviewer#3
- The authors introduced new method CGB barcodes on SARS-CoV and SARS-CoV-2 . Is it first report of CGB method?
- Did CGB method use in other viruses ? If there has reports on other viruses, please include it in the discussion part.
As indicated in the main text, this new method is partly based on the SWB method published by Hassanin et al. in 2022. However, the CGB barcodes represent a new method.
The method has not yet been used on other datasets, but we obviously plan to use it soon to study the evolution of other viruses.
- It is better to describe limitation of CGB method.
The limitations of phylogenetic, host and geographic CGB barcodes are already described in the section 4.4. Conclusion and perspectives.
- This study indicate host and geographic origins of SARS-CoV & SARS-CoV-2 (Sarbecovirus Lineage B). Why the authors does not includes other Sarbecovirus like as Bat SL-CoV-WIV1 RaTG13?
We tried to include all the diversity of Sarbecovirus genomes currently available in international databases. As indicated in section 2.1, “several genomes showing perfect identity or high nucleotide similarity (more than 99.9% of nucleotide identity) were published for pangolin sarbecoviruses from Guangxi (5 sequences), bat sarbecoviruses from Thailand (5 sequences), Cambodia (two sequences), etc. For these clusters, a single genome was retained for our analyses.”
Please note that RaTG13 is of course included in our dataset, as well as Rs3367, which shares 99.9% of nucleotide identity with Bat SL-CoV-WIV1 (Ge et al., 2013). See our supplementary Table S1 for the origin of the sequences used in this study.
- The author described that SARS-CoV-2 share ancestry with Rinolophus viruses from Yunnan, Lao and Cambodia. How about relation with other countries?
As shown in Figure 6, the contribution of Yunnan and northern Laos is much more important than other regions of China and other countries of SE Asia, such as Cambodia and Thailand.
- In this study, the authors described genome, host taxa, geographic regions sharing between SARS-CoV-2 and several bats or pangolin viruses within Asian countries. Kindly explain the reason why you focused on Asia countries.
Actually, we don’t understand the remark of reviewer#3 because we did not focus on Asian viruses. Our dataset includes all the diversity of Sarbecovirus genomes currently available in the international databases. However, these viruses are much more diversified in China and Southeast Asia and our results show that SARS-CoV originated from Yunnan and that SARS-CoV-2 originated from Yunnan or northern Laos.
Round 2
Reviewer 1 Report
Most revisions are OK